# Controllability governs the balance between Pavlovian and instrumental action selection

Hayley M. Dorfman [1]* & Samuel J. Gershman [1]

A Pavlovian bias to approach reward-predictive cues and avoid punishment-predictive cues can conflict with instrumentally-optimal actions. Here, we propose that the brain arbitrates between Pavlovian and instrumental control by inferring which is a better predictor of reward. The instrumental predictor is more flexible; it can learn values that depend on both stimuli and actions, whereas the Pavlovian predictor learns values that depend only on stimuli. The arbitration theory predicts that the Pavlovian predictor will be favored when rewards are relatively uncontrollable, because the additional flexibility of the instrumental predictor is not useful. Consistent with this hypothesis, we find that the Pavlovian approach bias is stronger under low control compared to high control contexts.

[1] Department of Psychology and Center for Brain Science, Harvard University, Northwest Lab Building, 52 Oxford Street, Cambridge, MA 02138, USA.
*email: hdorfman@g.harvard.edu

A long-standing distinction holds that a Pavlovian learning system controls behavioral responses based on stimulus-outcome relationships (independently of actions), whereas a separate instrumental learning system controls responses based on stimulus-action-outcome relationships. In violation of this strict dichotomy, Pavlovian processes are known to promote approach towards reward-predictive stimuli and avoidance of punishment-predictive stimuli[1], even when they produce maladaptive behavior[2]. For example, Hershberger[3] famously demonstrated that newborn chicks struggled to learn that they should walk away from a cup of food in order to obtain it. The chicks could not suppress their Pavlovian tendency to move toward the cup, which was rigged to move farther away as the chicks approached. Another example of this phenomenon, referred to as Pavlovian misbehavior, comes from studies of autoshaping, in which animals interact with a reward-predictive cue (e.g., pigeons will peck a keylight that precedes pellet delivery) despite the fact that these behaviors do not affect the reward outcome. If an omission contingency is then introduced, such that expression of these behaviors causes the reward to be withheld, animals will sometimes persist in performing the maladaptive behavior, a phenomenon known as negative automaintenance[4]. Humans also exhibit Pavlovian misbehavior in Go/No-Go tasks, erroneously acting in response to reward-predictive stimuli when they should withhold action, and erroneously withholding action in response to punishment-predictive stimuli when they should act[5,6].

The idea that instrumental and Pavlovian processes coexist and compete for control of behavior has been a long-standing fixture of associative learning theory[7–9], and more recently has been formalized within the framework of modern reinforcement learning theories[10]. These theories have typically assumed that instrumental and Pavlovian processes each provide action values, which are then linearly combined to produce composite action values that control behavior. A weighting parameter determines the degree of Pavlovian influence, and this parameter is fit to each participant in the experimental data set. In this paper, we argue that the weighting parameter is determined endogenously by an arbitration process, much like an influential proposal for the arbitration between model-based and model-free reinforcement learning strategies[11].

Our theory of arbitration is based on the idea that Pavlovian and instrumental processes can be understood as constituting different predictive models of reward (we will use the terms 'predictor' and 'model' interchangeably, except where we distinguish the brain's internal models of the environment from our models of the brain). The instrumental predictor learns reward expectations as a function of both stimuli and actions, whereas the Pavlovian predictor learns reward expectations as a function only of stimuli. Thus, the instrumental predictor is strictly more complex than the Pavlovian predictor: it can capture any pattern that the Pavlovian predictor can capture, as well as patterns that the Pavlovian predictor cannot capture. The cost of this flexibility is that the instrumental predictor can also overfit on a finite data set, which means that it will generalize poorly due to fitting noise. The basic problem of arbitration is thus to negotiate a balance between capturing the patterns in the data (favoring the more complex instrumental predictor) and avoiding overfitting (favoring the less complex Pavlovian predictor).

Bayesian model averaging elegantly resolves this problem by weighting each predictor's output by the posterior probability of the predictor given the data. The posterior will tend to favor predictors of intermediate complexity, due to what is known as Bayesian Occam's razor[12]. We can think of each predictive model as 'betting' on observing particular data sets (Fig. 1a). Simple models concentrate their bets on a relatively small number of data sets, whereas complex models distribute their bets across a larger number of data sets. If a simple model accurately predicts a particular data set, it is rewarded more than a complex model, because it bet more on that data set. If the model is too simple (i.e., its bets are too narrowly concentrated), it will fail to predict the observed data.

Another perspective on the same idea comes from the bias-variance trade-off[13–15]. Any predictor's generalization error (i.e., how poorly it predicts new data after learning from a finite amount of training data) can be decomposed into the sum of three components: squared bias, variance, and irreducible error. Bias is the systematic error incurred by adopting an overly simple model that cannot adequately capture the underlying regularities in the data. Variance is the random error incurred by adopting an overly complex model, which causes the model to overfit random noise in the training data. The irreducible error arises from the inherent stochasticity of the data-generating process, which is independent of the predictor. Bias can be reduced by increasing model complexity, but at the cost of increasing variance. Optimal generalization error is achieved at an intermediate level of complexity where the sum of squared bias and variance (i.e., the reducible error) is minimal (Fig. 1b). The bias-variance trade-off is closely connected to the Bayesian model averaging perspective, because predictive models with higher posterior probability will tend to have lower generalization error[16].

Applying these ideas to arbitration between Pavlovian and instrumental control, a key determinant of the optimal model complexity is controllability of reward[17,18]. If rewards are uncontrollable (actions do not affect reward rate), then the simpler Pavlovian predictor will be favored by the posterior, because the additional complexity of the instrumental predictor is not justified relative to the penalty imposed by the Bayesian Occam's razor. Only when rewards are sufficiently controllable, or once sufficient data have been observed, will the instrumental predictor be favored (asymptotically, the instrumental predictor will always be favored, because the risk of overfitting noise disappears as the data set becomes large).

We test the predictions of the Bayesian arbitration model by manipulating reward controllability in two Go/No-Go experiments, using the Pavlovian Go bias observed in previous experiments[19,20] as an index of Pavlovian control. As a complementary window into the arbitration process, we also explore how controllability affects the bias-variance trade-off.

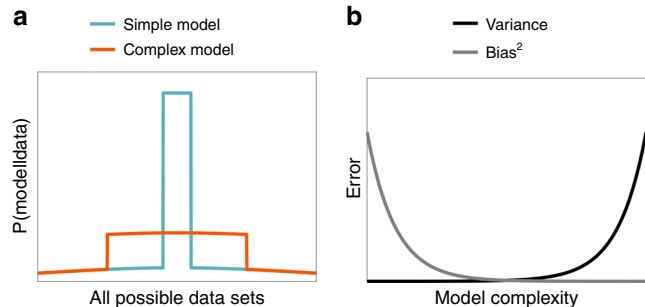

**Fig. 1 Two perspectives on model complexity. a** Bayesian Occam's razor. Complex models distribute their probability mass across many different data sets, and thus get less credit for observing any particular data set, whereas simple models concentrate their probability mass on a small number of data sets, and thus, get relatively more credit when those data sets are observed. **b** As model complexity increases, generalization error due to bias decreases, while generalization due to variance increases.

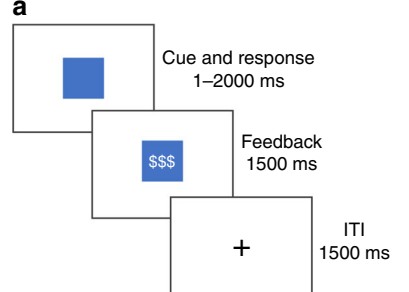

| | Go | No-go |
|---|---|---|
| Go-to-win | 0.75 | 0.25 |
| No-go-to-win | 0.25 | 0.75 |
| Low control decoy | 0.5 | 0.5 |
| Low control decoy | 0.8 | 0.2 |

**Fig. 2 Behavioral task details. a** Participants viewed a colored shape cue (for up to 2 s) and had to decide whether to press the space bar (Go) or refrain from pressing the space bar (No-Go). They then received feedback (1.5 s) denoted by dollar signs (reward) or a rectangular cue (neutral). Participants were instructed that they would receive a small amount of real bonus money for each rewarded outcome, and no bonus money for each neutral outcome. Feedback was followed by an inter-trial-interval (ITI, 1.5 s). **b** Reward contingencies for each trial type (Go-to-Win; No-Go-to-Win; Decoy) and action type (Go; No-Go). Task condition was manipulated either between (Experiment 1) or within (Experiment 2) participants. Across both experiments, the Low Control Decoy only appeared in the Low Control condition, and the High Control Decoy only appeared in the High Control condition.

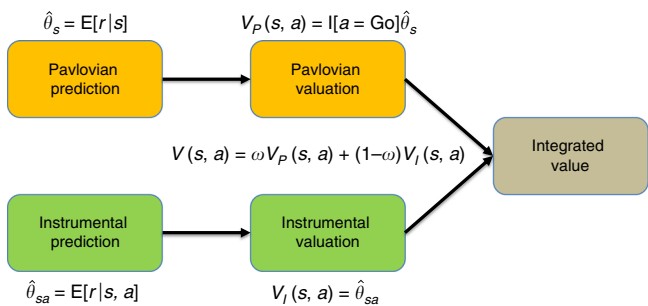

**Fig. 3 Information processing architecture.** Pavlovian and instrumental prediction and valuation combine into a single value. This integrated value includes a weighting parameter ($w$) that represents the evidence for the uncontrollable environment (i.e., in favor of the Pavlovian predictor). See Methods for technical details.

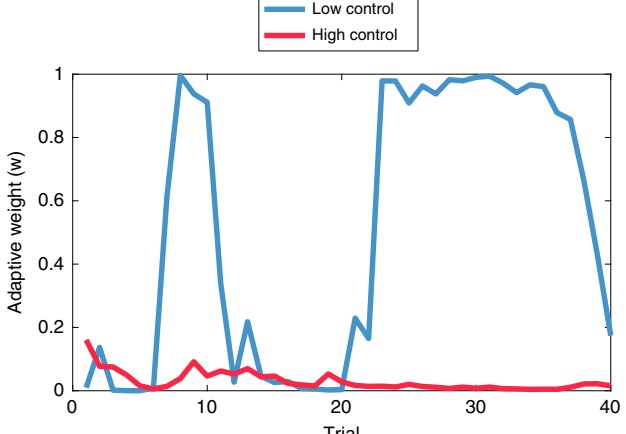

**Fig. 4 Weight dynamics for one participant.** The adaptive model demonstrates a greater reliance on the Pavlovian system in the Low Control condition compared to the High Control condition.

## Results

**Behavior and modeling**. We describe the two experiments together because they are very similar in structure (Fig. 2). Experiment 1 manipulated reward controllability between participants, whereas Experiment 2 manipulated it within participants.

To investigate the extent to which participants relied on Pavlovian control, we measured their Go bias, defined as the difference in accuracy on Go-to-Win and No-Go-to-Win trials (see Supplementary Fig. 1 for the disaggregated data). Under purely instrumental control, the Go bias should be 0; hence values greater then 0 indicate the influence of Pavlovian control.

We developed two models of behavior on this task (see Methods for details). Both models consist of two sub-components; a Pavlovian learning system and an instrumental learning system (Fig. 3). The Pavlovian system acquires reward expectations for each stimulus, and converts these expectations into action values by promoting Go responses to cues in proportion to their expected reward. The instrumental system acquires reward expectations for each stimulus-action combination, and converts these into action values by promoting Go responses to stimuli in proportion to their expected reward for Go relative to No-Go. The learning rules for both systems are the same.

The Pavlovian and instrumental values are integrated linearly according to a weighting parameter. The two models differ in terms of how the weighting parameter changes as a function of experience. In the fixed model, the weighting parameter is held constant, treated as a free parameter that we fit to behavior. In the adaptive model, the weighting parameter is updated after each trial based on the relative predictive ability of each system. Thus, the weight is not a free parameter in the adaptive model, but is instead determined endogenously by each participant's experience in the task.

Figure 4 shows the time series of the adaptive Pavlovian weight for the model fitted to one participant, demonstrating the prediction that low control should tend to produce a higher Pavlovian weight ($w$), which will in turn cause a stronger Go bias.

Consistent with the model predictions, participants across both experiments showed an increased Go bias in the Low Control (LC) condition compared to the High Control (HC) condition (Experiment 1: $t(183) = 2.06$, $p < 0.05$, $d = 0.31$; Experiment 2: $t(128) = 2.06$, $p < 0.05$, $d = 0.18$; by two-sample $t$-test; Fig. 5). The adaptive model provided a quantitatively superior account relative to the fixed model, as assessed by random effects Bayesian model comparison[21]. Specifically, we calculated the protected exceedance probability (PXP), the probability that a particular model is more frequent in the population than all other models under consideration, taking into account the possibility that some differences in model evidence are due to chance. For both experiments, the PXP favoring the adaptive model was >0.99.

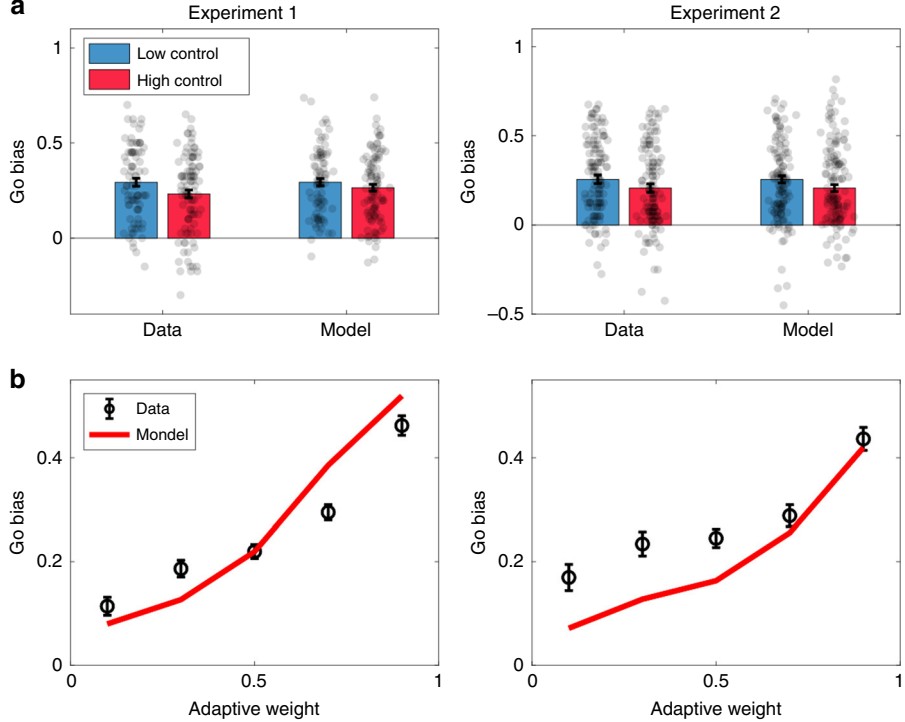

**Fig. 5 Go bias for low and high control conditions across experiments. a** Go bias for low and high control conditions in Experiment 1 (left) and Experiment 2 (right). **b** The adaptive model captures within-participant variability in Go bias, plotted as a function of Pavlovian weight (*w*) quantile for Experiment 1 (left) and Experiment 2 (right). Error bars show standard error of the mean.

To verify the quantitative accuracy of the adaptive model, we plotted the Go bias as a function of weight quantile (Fig. 5), finding a close fit between model and data (for both experiments, the *t*-test comparing the Go bias for the lowest and highest quantiles was significant; Experiment 1: $t(177) = 12.30$, $p < 0.0001$, $d = 0.92$; Experiment 2: $t(54) = 6.16$, $p < 0.0001$, $d = 0.83$), apart from a slight deviation in Experiment 2 for the lowest weight quantiles. Importantly, the quantiles were computed within participants, demonstrating that the model can capture variations in Pavlovian control over the course of a single experimental session.

The timeseries of weights generated by the adaptive model is, on average, correlated with the parameter estimates obtained from fitting the fixed model (Experiment 1: $r = 0.51$, $p < 0.0001$; Experiment 2: $r = 0.63$, $p < 0.0001$; by correlation). This demonstrates that the adaptive model's average behavior produces behavior similar to that predicted by earlier models using fixed weights[19,20] but with the weight determined endogenously rather than fit as a free parameter.

We also tested the prediction that the Go bias should diminish over the course of training, and eventually disappear, as can be seen in the simulations (Fig. 4). Consistent with this prediction, the Go bias in both experiments declined over the course of trials, roughly exponentially fast (Fig. 6). Specifically, we regressed the Go bias against the log-transformed trial number and then carried out *t*-tests on the regression coefficient, finding a significant negative effect for Experiment 1 ($t(184) = 2.93$, $p < 0.005$, $d = 0.22$) and Experiment 2 ($t(128) = 5.55$, $p < 0.0001$, $d = 0.49$).

**Analysis of bias and variance**. We also examined the effect of controllability on the bias-variance trade-off (Fig. 7). Because controllability favors the more complex instrumental model, we hypothesized that the HC condition would produce lower bias

and higher variance (note that this bias should not be confused with the Pavlovian Go bias; see Supplementary Fig. 3 for model simulations of bias and variance). This prediction was partially confirmed in both Experiment 1 (bias: $t(183) = 2.06$, $p < 0.05$, $d = 0.31$; variance: $t(183) = 1.69$, $p = 0.09$, $d = 0.25$) and Experiment 2 (bias: $t(128) = 2.07$, $p < 0.05$, $d = 0.18$; variance: $t(128) = 2.37$, $p < 0.05$, $d = 0.21$; by *t*-test).

## Discussion

Taken together, our experimental data provide evidence consistent with a Bayesian model averaging theory of Pavlovian-instrumental arbitration. Our key finding was that the Pavlovian Go bias was stronger under conditions of low reward controllability, consistent with our model's prediction. Analyses in terms of the bias-variance trade-off supported the same conclusion: low controllability favors the simpler Pavlovian predictor, leading to high bias and low variance.

Our results cannot be explained by a non-specific Go bias, whereby Go responses are rewarded more in the High control condition, inducing an overall tendency to produce Go responses. This would in fact predict the opposite effect (stronger Go bias under high reward controllability), contrary to our experimental findings. Even stronger evidence against a non-specific Go bias would be provided by a version of the experiment in which participants must make Go/No-Go responses to avoid punishment.

The idea that Pavlovian-instrumental interactions are governed by probabilistic inference joins a number of related ideas in the theories of reinforcement learning. Most relevantly, Daw and colleagues suggested that arbitration between model-based and model-free control was determined by Bayesian arbitration[11], but they did not address Pavlovian-instrumental interactions. A number of earlier theories argued that certain reinforcement learning behaviors could be understood as arising from a model comparison process[22–25]. However, to our knowledge, ours is the

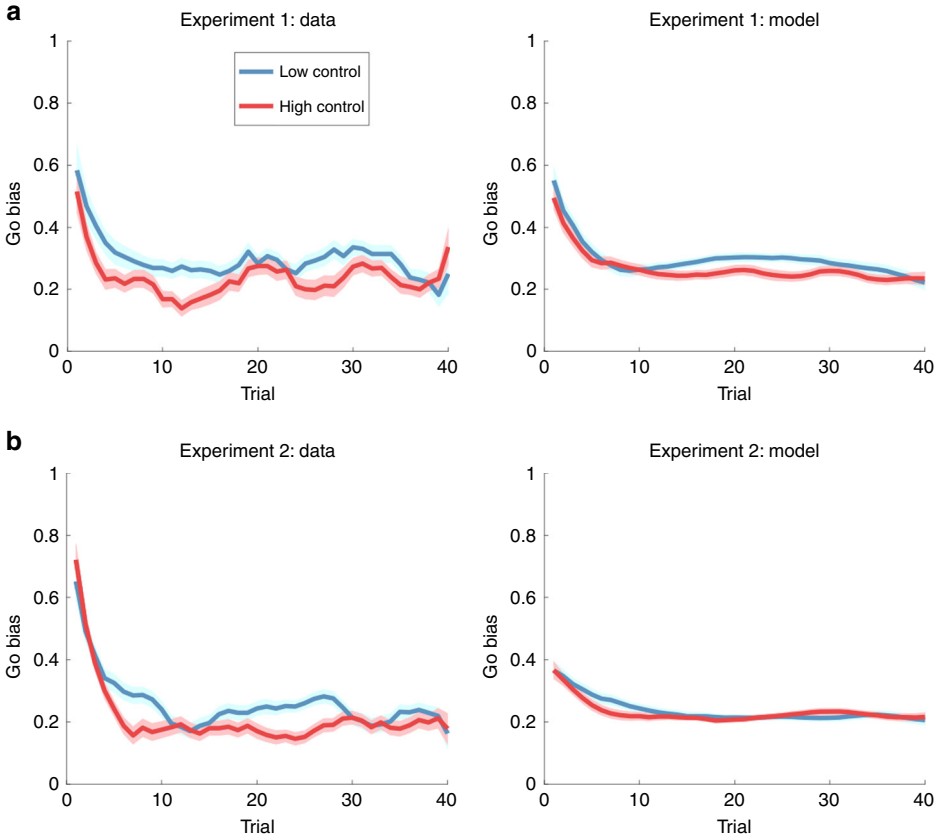

**Fig. 6 Go bias as a function of trial.** Go-bias across trials for **a** Experiment 1 and **b** Experiment 2. The data plots have been smoothed with a 5-trial moving average. Error bars show standard error of the mean.

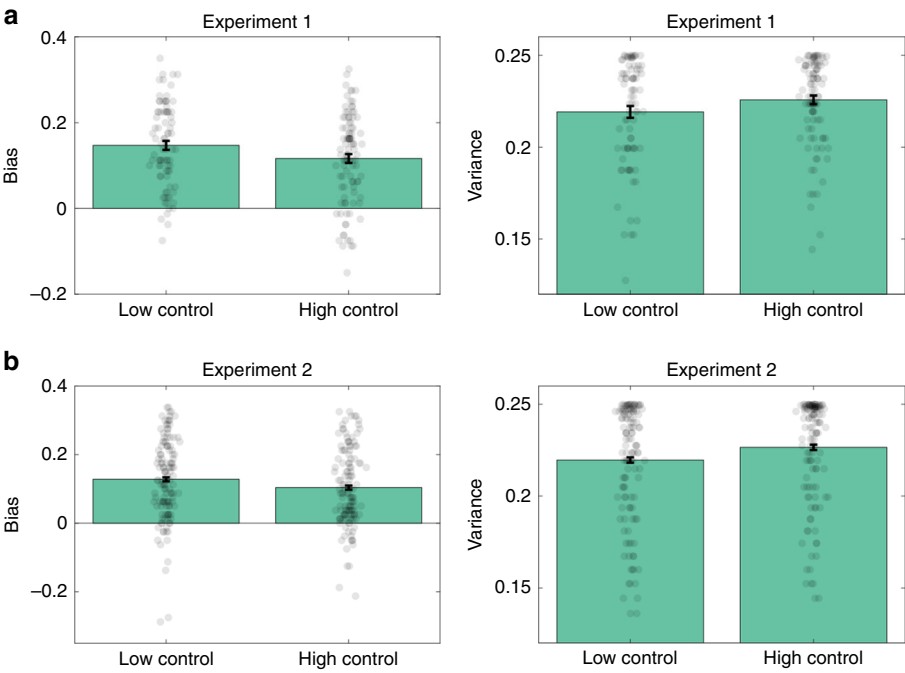

**Fig. 7 Bias and variance across conditions and experiments.** Bias and variance of choice behavior for **a** Experiment 1 and **b** Experiment 2. Error bars show standard error of the mean.

first account that directly addresses Pavlovian-instrumental interactions in terms of model comparison/averaging.

Recent work by Swart and colleagues complicates this picture by showing that the Go bias is at least partially accounted for by instrumental learning biases[26]. In particular, participants in their study tended to learn more quickly from rewarded Go trials compared to rewarded No-Go trials, whereas they learned more slowly from punished No-Go trials compared to punished Go trials. This instrumental learning bias causes Go responses to appear more attractive when correct actions yield reward, and less attractive when correct actions yield avoidance of punishment. This phenomenon is not accounted for by our modeling framework.

Our results suggest several directions for future work. First, we have only studied the dynamics of the Pavlovian go bias for rewards; earlier work suggests that we should find a symmetric pattern for punishments, with a stronger No-Go bias under low controllability[6,27]. Second, neuroimaging could be used to identify the neural correlates of arbitration. If our account is correct, we would expect to see a signal in the brain that encodes the dynamically changing weight parameter. Third, an open theoretical task will be to generalize the model to explain other forms of Pavlovian-instrumental interactions, such as negative auto-maintenance and Pavlovian-instrumental transfer.

More broadly, our findings are consistent with the idea that agency is one factor that can mediate the trade-off between learning processes, which has important implications for understanding psychopathology. For example, many studies in both humans and animals have shown that controllability (or lack thereof) influences future instrumental responding. Learned helplessness, where the experience of uncontrollable punishments leads to diminished instrumental learning (for example, failure to learn to escape an electric shock[28], is hypothesized to be a model of, and has been linked to, symptoms of depression and anxiety[29]. Although the learned helplessness literature has focused on uncontrollable punishments, there is also evidence that individuals with depression are less likely to experience illusions of control with rewards[30]. The idea that inferences about controllability underlie learned helplessness has been incorporated into formal Bayesian models that share some properties with the model proposed in this paper[31]. In addition, recent work has shown that outcome controllability manipulations can induce learned helplessness in humans, and also enhance Pavlovian biases in a reinforcement learning context[32].

In conclusion, we have shown how the framework of Bayesian model averaging can shed light on the cognitive mechanisms underlying Pavlovian misbehavior. Although the simple model studied in this paper is not a comprehensive theory of Pavlovian-instrumental interactions, it points towards one mechanism that is likely to play an important role in future, more comprehensive theories.

## Methods

**Participants**. We recruited two independent samples of adults from Amazon Mechanical Turk (Experiment 1: N = 271, Experiment 2: N = 183). The sample sizes were chosen in order to exceed sample sizes from previous, similar work[5,6,19]. Participants for Experiment 2 were recruited from an existing pool of Amazon Mechanical Turk workers. These workers have completed previous experiments for our lab and expressed interest in being re-contacted for additional study opportunities. Participants were excluded for inaccuracy. Specifically, if participants made the incorrect action (either a button press for a No-Go trial, or the absence of a button press for a Go trial) for ≥50% of all trials, they were excluded from analyses. We also excluded any participants that performed <30% on any one condition. This left a total of 185 accurate participants for Experiment 1 and 129 accurate participants for Experiment 2 (see Supplementary Fig. 2 for experimental results without participant exclusions). The Harvard University Committee on the Use of Humans Subjects approved the experimental procedures and participants provided informed consent prior to beginning the study.

**Procedure**. Participants completed a modified Go/No-Go paradigm where they made a decision on each trial to either take or avoid an action in response to a stimulus to receive reward[6,20]. Participants viewed a single colored square on each trial and had to learn the appropriate response for each square. There was a different correct response and reward probability combination for each shape: One square was a Go stimulus, where a spacebar press was rewarded 75% of the time, one square was a No-Go stimulus, where the absence of a button press was rewarded 75% of the time, and the third square was a Decoy stimulus, where a spacebar press was rewarded with a particular probability, which was manipulated based on experimental condition. In the Low control (LC) condition, the Decoy was rewarded 50% of the time, and in the High control (HC) condition—the Decoy was rewarded 80% of the time. Our task differed from previous Go/No-Go tasks in that it did not include any punishment conditions. Rewarded outcomes were represented with dollar signs, and unrewarded outcomes were represented with a neutral (white rectangle) cue. Participants were told that they would receive a small amount of real bonus money for each reward outcome, and their total bonus was summed and disclosed at the end of the experiment.

In Experiment 1, participants were randomly assigned to one decoy condition (LC or HC), so that each participant was exposed to three different stimuli (Go-to-Win, No-Go-to-Win, and either LC or HC). The experiment consisted of 120 trials, 40 trials for each type of stimulus, randomly interleaved. In Experiment 2, each participant experienced both decoy conditions in separate blocks, the order of which was randomized. The experiment consisted of 240 trials, 120 for each block, with 40 trials for each stimulus within a block. The experiment was coded in jsPsych, version 6.0.5[33].

**Computational model**. On each trial of the task, the participant must take an action ($a$) in response to a stimulus ($s$) in order to receive a reward ($r$). The problem facing the participant is to determine whether they are acting in an environment where outcomes are controllable (instrumental) or uncontrollable (Pavlovian).

Each model has a set of parameters $\theta$ that must be learned. The parameters for the uncontrollable model are indexed only by the stimulus ($\theta_s$), whereas the parameters for the controllable model are indexed by both the stimulus and action ($\theta_{sa}$). We will walk through the learning equations for the uncontrollable model, but the idea is essentially the same for the controllable model (see Supplementary Methods for complete derivations).

The posterior over parameters given data $\mathcal{D}$ (the history of stimuli, actions and rewards) and environment $m \in$ {controllable, uncontrollable} is stipulated by Bayes' rule:

$$P(\theta|\mathcal{D}, m) \propto P(\mathcal{D}|\theta, m)P(\theta|m) \tag{1}$$

where $P(\mathcal{D}|\theta, m)$ is the likelihood of the data given hypothetical parameter values $\theta$, and $P(\theta/m)$ is the prior probability of those parameter values. In the context of our task, where rewards are binary, $\theta_s = \mathbb{E}[r|s]$ corresponds to the mean of a stimulus-specific Bernoulli distribution. When $P(\theta_s)$ is a $Beta\left(\theta_0 \frac{\eta_0}{2}, (1 - \theta_0)\frac{\eta_0}{2}\right)$ distribution, the posterior mean $\hat{\theta}_s$ (which is also the posterior predictive mean for reward) is initialized to $\theta_0$ and updated according to:

$$\Delta\widehat{\theta}_s = \eta_s^{-1}\delta \tag{2}$$

where $\delta$ is the reward prediction error ($r - \hat{\theta}_s$), and $\eta_s^{-1}$ is the learning rate with counter $\eta_s$ initialized to $\eta_0$ and incremented by 1 every time stimulus $s$ is encountered (in the controllable model, $\eta$ is indexed by both $s$ and $a$). Intuitively, $\theta_0$ corresponds to the prior mean (the reward expectation before any observations), and $\eta_0$ corresponds to the prior confidence (how much deviation from the prior mean the agent expects).

Because the true environment is unknown, it must be inferred, which can be done using another application of Bayes' rule:

$$P(m|\mathcal{D}) \propto P(\mathcal{D}|m)P(m) \tag{3}$$

where

$$P(\mathcal{D}|m) = \int P(\mathcal{D}|\theta, m)P(\theta)d\theta \tag{4}$$

is the marginal likelihood. The posterior can be updated in closed form. For clarity we adopt a log-odds convention, with the prior log-odds given by:

$$L_0 = \log\frac{P(\text{uncontrollable})}{P(\text{controllable})} \tag{5}$$

The posterior log odds are initialized to $L_0$ and updated according to:

$$\Delta L = r\log\frac{\hat{\theta}_s}{\hat{\theta}_{sa}} + (1 - r)\log\frac{1 - \hat{\theta}_s}{1 - \hat{\theta}_{sa}} \tag{6}$$

Finally, we need to specify how each model maps reward predictions onto action values. For the instrumental model, we assume that action values simply correspond to the expected reward for a particular state-action pair: $V_I(s, a) = \hat{\theta}_{sa}$. For the Pavlovian model, we assume that the action value is equal to $V_P(s, a) = 0$ for $a =$ No-Go and $V_P(s, a) = \hat{\theta}_s$ for $a =$ Go. This assumption follows from the influential idea that Pavlovian reward expectations invigorate action[5]. To combine

the two action values into a single integrated value for action selection, we weight each model's value by its corresponding posterior probability:

$$V(s,a) = wV_P(s,a) + (1-w)V_I(s,a), \quad (7)$$

where

$$w = P(m = \text{uncontrollable}|\mathcal{D}) = \frac{1}{1+e^{-L}} \quad (8)$$

is the posterior probability of the uncontrollable environment.

To allow for stochasticity of behavior, we model the agent's action selection according to a softmax, where $\beta$ is an inverse temperature parameter controlling the level of choice stochasticity:

$$P(a|s) = \frac{\exp[\beta V(s,a)]}{\sum_{a'}\exp[\beta V(s,a')]} \quad (9)$$

The model outlined above, which we will refer to as the adaptive model, updates the weighting parameter from trial-to-trial based on the relative predictive accuracy between the two controllers. We also fit a comparison model, which instead fits the weighting term as a free parameter. We refer to this comparison model as the fixed model. The models share the same underlying information processing architecture (Fig. 3) but differ in whether $w$ is set exogenously (in the case of the fixed model) or endogenously (in the case of the adaptive model).

We fit each model's free parameters using maximum likelihood estimation. The adaptive model had five free parameters: the inverse temperature $\beta$, and the parameters of the prior ($\theta_0, \eta_0$) for each environment (High or Low Control). We also considered a model in which $L_0$ was fit as a free parameter, but model comparison indicated that fixing $L_0 = 0.5$ had greater support in our data sets. The fixed model had six free parameters: the same five as the adaptive model, plus the weighting parameter $w$. Average parameter estimates are reported in Supplementary Table 1.

**Bias-variance analysis**. To assess how controllability affects the bias-variance trade-off, we calculated these quantities for each participant as follows:

$$bias = \frac{1}{T}\sum_{t=1}^{T}\mathbb{I}[a_t = Go] - \mathbb{I}[a_t^* = Go] \quad (10)$$

$$variance = \frac{1}{T}\sum_{t=1}^{T}(\mathbb{I}[a_t = Go] - \bar{a}_t)^2 \quad (11)$$

where $a_t$ is the chosen action on trial $t$, $a_t^*$ is the optimal action, $\bar{a}_t = \frac{1}{T}\sum_{t=1}^{T}\mathbb{I}[a_t = Go]$, $T$ is the number of trials (note that the optimal action is not defined for the Low control decoy), and $\mathbb{I}[\cdot] = 1$ when its argument is true, and 0 otherwise.

Intuitively, this bias measures how much a participant's actions deviate from the optimal policy. A bias of 0 indicates that the participant always follows the optimal policy. Positive values indicate that the participant is responding Go more frequently than optimal. The variance measures how much a participant's actions deviate from their average policy. A variance of 0 indicates that the participant always gives the same response.

**Reporting summary**. Further information on research design is available in the Nature Research Reporting Summary linked to this article.

## Data availability
The data that support the findings of this study are available at: https://github.com/sjgershm/GoNogo-control.

## Code availability
Code to produce computational model results and plots is available at: https://github.com/sjgershm/GoNogo-control and code for the experiment is available at: https://github.com/hayleydorfman/pavlovian-instrumental-arbitration.

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

## Acknowledgements
We would like to thank Rebecca Hao for help with the initial setup for this study. This work was supported by the Office of Naval Research (N00014-17-1-2984) and the Alfred P. Sloan Foundation.

## Author contributions
S.J.G. designed the study; H.M.D. collected the data; S.J.G. and H.M.D. analyzed the data; S.J.G. designed the computational models; S.J.G. and H.M.D. wrote the manuscript.

## Competing interests

The authors declare no competing interests.
