## [Peer Review File · Nature Communications]

Reviewers' Comments:

Reviewer #1:

Remarks to the Author:

In this manuscript, the authors present data and computational analysis to show that the extent to which human participants show a Pavlovian bias in action selection differs between two contexts of varying controllability. The authors hypothesised that Pavlovian control would overtake instrumental control in contexts where the link between action and rewards was weaker (low controllability) than in contexts where this link was stronger (high controllability). They put forward a Bayesian model in which they assume that the brain arbitrates between Pavlovian values (state values) and instrumental values (state action values) by means of a sigmoid function of the ratio of the posteriors of the model evidence (Pavlovian vs instrumental) given the data, that is how well each model fits the observed pattern of rewards. In contexts where some rewards are randomly obtained the ratio favours the Pavlovian model and whenever there are some action contingencies, the instrumental model will eventually take over. The authors also present empirical data that supports their theoretical hypothesis.

In general, I think that the manuscript is very well written and it was a pleasure read. The hypothesis and theoretical work supporting the manuscript is novel, timing and very strongly grounded. Similarly, the empirical data seems to support the authors' claims. However, I would need further analysis of the empirical data to be convinced that this is the case. I also have some comments that may improve the presentation and discussion of the results.

1) The authors collapse the data of the 3 conditions (go to win, no go to win, and controllability condition) and never show them separately. I think that it is of vital importance that the authors can show that the performance on the go to win and no go to win condition is affected by varying the nature of the controllability condition. Otherwise, the changes in go bias could be trivially attributed to changes in performance in the controllability condition only. Related to that, one could expect that the posterior odds are updated for each condition separately

2) Regarding the empirical data, the authors only present the overall go bias and mainly disregard learning effects (only coarsely presented for experiment 1 and collapsing across controllability conditions). This is puzzling considering that the authors used a learning task and that their model makes predictions about the learning curves. Although learning effects may be more difficult to test in experiment 2 because of the blocked design, the authors should still present these data for both experiments.

Related to the previous point, in figure 5, the authors verify the quantitative accuracy of the adaptive model by binning action bias as a function of weight quantile. As the weight is provided by the model they are trying to test, this analysis seems somehow circular. Wouldn't it be better to use learning curves instead?

3) The presentation of the model would benefit from further development and clarifications to ensure that a wider audience can easily understand it. This applies to the whole modelling section but especially to the section about bias-variance analysis.

4) The authors assume that the instrumental learning controller is unbiased. Recent computational and experimental work suggest that biased instrumental learning mechanisms that reflect dopamine dependent learning mechanisms in the basal ganglia (Swart et al 2017, eLife). The authors should consider this work and its implications for their model in the discussion.

5) reference 28 appears to be incomplete

Reviewer #2:

Remarks to the Author:

In this paper, the authors show that the controllability of an environment determines the balance between Pavlovian and instrumental actions. Controllability is minimum when the chance of reward is 50-50 given actions. The authors further propose a computational model which is aimed to explain how controllability modulates the balance between the Pavlovian and instrumental processes. The task is relatively similar to the previous ones, and the idea of controllability of reward is previously introduced (e.g. in 17). The novel part is then the computational modelling approach to combine Pavlovian and instrumental processes. I find the ideas overall interesting, however, I have the following concerns.

- If what is determining the balance is controllability, then we expect to see the same effects when in HC condition the probabilities are 20(GO)-80(No-GO)—the controllability in this condition is the same as the condition reported in the paper and therefore they should provide same Go Bias. I find this important to test this condition in order to confirm that the effects are purely due to controllability and not due to the imbalance in the dataset in HC trials.

- I found it kind of odd that the weight parameter is determined based on which model provides the best fit to data. The adaptive model basically gives more weight to the system which can predict the data better. I think it is always the case that when we give more weight to the system which explained the data better (in a hybrid structure), the overall fit will be better than a hybrid structure which gives a fixed weight to each system. As such, the better fit of the adaptive model (compared to the fixed model) does not provide new information. In order to avoid this circular argument, the weight parameter I think should be determined by another measure, e.g., directly through controllability, instead of indirectly through the model fits.

- From Figure 4 left panel, it seems that the differences between model fits is driven by the learning process of the subjects, and it is not purely reflecting the controllability of the environment. That is, in both conditions the instrumental model is able to provide the better fit to data but it needs more learning. It then bears the question of whether the transient effects that differentiate controllability (early trials in Figure 4 left) are rooted in the Pavlovian and instrumental nature of the models (stimulus vs stimulus-action values tracking) or the transient effects are purely due to the specific model parameterisation used here. For example, would we see a similar pattern if a fixed learning rate is used instead of the one in Line 188? What if the instrumental system is parametrized to track the difference between values? Please clarify the generality of the results.

- Bias-variance terms in line 248 – The values of these two quantities shown in Fig 7 are very small. I suspect a division by the number of trials is required in these equations. The relationship between bias and variance here and the trade-off between the bias and variance of an estimator is not very clear. With two actions if the subject is biased towards one of them then naturally the variance will be lower. It would be useful if the authors expand what exactly is going to be estimated here (the optimal action?) and whether the variance and bias generated by the two models are inline the assumptions.

Minor comments:

- Please add the derivations of the equations, e.g., the equation below line 200 to supplementary materials (and please number the equations).

- How the optimal action (a^*_n) was chosen in the 50-50 conditions (line 248)?
- Please show the performance of the subjects.
- Equations below line 248, it would be better to replace the summation index with 't'.
- It might not be accurate to talk about "actions values" within a Pavlovian system (line 49), as learning the value of actions (as opposed to learning the value of stimuli) is more related to the instrumental processes.
- Please show Figure 4 (left) for each trial type.

Reviewer #3:

Remarks to the Author:

OVERALL AIM OF PAPER

Dorfman and Gershman revisit the hypothesis that the degree of Pavlovian influence on behavior is fixed and propose that the weight on Pavlovian versus instrumental control is instead determined endogenously by an arbitration process in a dynamic manner that depends on a key meta-parameter estimate of the environment: its controllability.

MORE SPECIFIC SUMMARY

The authors approach the problem of arbitration by considering it to reflect Bayesian model averaging and a bias-variance trade-off (between error induced by overly simple models and that induced by overly complex models). To test predictions of the Bayesian arbitration model and the bias-variance trade-off, they manipulated reward controllability, between subjects (exp 1) and within subjects (exp 2), in two Go/No-Go experiments, revealing the Pavlovian go bias. This was done by exposing people to randomized sequences of 3 stimuli, 1 stimulus per trial, with a button press for each of 3 stimuli being associated with either 75%, 25% and, either 50% (for the uncontrollable condition) or 80% (for the controllable condition) reward probability. The GO bias (difference in %go for 75% and 25% stimuli was greater in the controllable condition (if the 75 and 25% stimuli were interleaved with the 80% stimulus) than in the uncontrollable condition (if the 75 and 25% stimuli were interleaved with the 50% stimulus). The data were best captured by an adaptive Bayesian model averaging model where a weighting parameter (indexing a trial-by-trial estimate of the uncontrollability of the environment) was estimated based on the relative predictive accuracy of the Pavlovian and instrumental controllers.

GENERAL EVALUATION

This is a well written and concise paper that addresses a timely topic, providing an original novel formal model of arbitration between Pavlovian and instrumental control. I do have the following comments that should be addressed:

MAJOR COMMENTS

It is stated that most previous work assumes the Pavlovian bias to be a fixed trait. This is not the case. Consider the extensive attempts to assess the effects of psychoactive drugs, but also other states such as stress, threat of shock and 'cognitive control' on precisely this bias.

The number of excluded subjects (e.g. 91 out of 189 in Exp1) is enormous, even for an MTurk study. What is going on? Can their behavior be characterized (e.g. in terms of its consistency with an exclusive Pavlovian bias strategy, or a simple winstay/losheshift strategy) and reported as such?

I was puzzled by the lack of a punishment manipulation for 2 reasons:

First, doesn't this imply that any of the effects might reflect modulation of a nonspecific Go bias rather than specifically a Pavlovian bias? And might this represent a spillover from a generally increased go

bias for the 80% stimulus? Second, the discussion highlights implications of the results for depression and learned helplessness, which concerns the aversive rather than appetitive domain.

Am I completely confused or is it the adaptive and not the fixed model that has 6 parameters, including the weighting parameter? (page 7)

Can the proportion of Go responses be plotted separately for the 75% go and the 25% nogo stimuli? This would enable the reader to assess whether the 80% stimulus reduced Go responding for the 25% nogo stimulus or increased it for the 75% go stimulus. Might the uncontrollable context elicit a less specific 'inertia' rather than a 'Pavlovian go bias'?

Can the estimated Pavlovian weight parameter be plotted across trials (as in the simulation Fig 4)? In fact, does it make sense to report all (predicted and estimated) parameter distributions?

Perhaps some bits of the paper are too concise.

(i) The reader would benefit from a clearer description of Pavlovian vs instrumental control, with reference to classical and operant conditioning.

(ii) Moreover, the authors could help the non-Bayesian-expert reader (like me) by making more intuitive earlier in the methods how to think about the mechanism by which environmental (un)controllability is inferred and how the fixed model differs conceptually from the adaptive model.

(iii) I am missing details of the simulation procedure. Is it the adaptive model that is simulated?

(iv) Figure 6 plots the Go bias as a function of time. Is this from the low control condition? What about the high control condition? Why not present data from the same analysis also for the within-subj exp and note the order issue. Here wouldn't a plot of the evolution of the weight parameter make sense?

It is unclear what is the added explanatory value of the bias-variance trade-off analysis. In fact, this compounds another question I had with regard to the statement that the current data provide evidence for a Bayesian model averaging model. What is the explanatory value of this framework if the Bayesian model averaging approach in the absence of evidence that it does a better job than alternative non-Bayesian approaches?

Computational Cognitive Neuroscience Lab
Harvard University
52 Oxford Street
Cambridge, MA 02138

Reviewer #1

1) The authors collapse the data of the 3 conditions (go to win, no go to win, and controllability condition) and never show them separately. I think that it is of vital importance that the authors can show that the performance on the go to win and no go to win condition is affected by varying the nature of the controllability condition. Otherwise, the changes in go bias could be trivially attributed to changes in performance in the controllability condition only. Related to that, one could expect that the posterior odds are updated for each condition separately.

The go bias is, by our operational definition, a difference in performance between go-to-win and no-go-to-win. We thus don't include the behavior for the decoy trials in any of our analyses of the go bias. However, we now include a new figure in the supplement (Fig S1) showing the behavior disaggregated across all conditions and stimuli.

If it were the case that the posterior over controllability was updated for each trial type separately, then we would not expect to see a difference in go bias across conditions. (We assume the reviewer is referring to "trials" not "conditions" here, and that "controllability condition" refers to the decoy stimulus. The posterior odds would of course be updated separately for each condition in Experiment 1, since each subject was only in one condition.)

2) Regarding the empirical data, the authors only present the overall go bias and mainly disregard learning effects (only coarsely presented for experiment 1 and collapsing across controllability conditions). This is puzzling considering that the authors used a learning task and that their model makes predictions about the learning curves. Although learning effects may be more difficult to test in experiment 2 because of the blocked design, the authors should still present these data for both experiments.

Thanks for this suggestion. As mentioned above, we now include Fig. S1, which shows the disaggregated data. We also have revised our figure showing the early vs. late comparison (Fig. 6) to include more detailed information from both experiments.

Related to the previous point, in figure 5, the authors verify the quantitative accuracy of the adaptive model by binning action bias as a function of weight quantile. As the weight is provided by the model they are trying to test, this analysis seems somehow circular. Wouldn't it be better to use learning curves instead?

We think of this as a predictive check: we want to visually confirm that the model is capturing the data. It's possible to fit a model that fails to capture the data, and we wouldn't know that without generating plots like this. It would only be circular if we were to claim on the basis of this plot that our model does better than another model. For such claims, we appeal to formal model comparison.

3) The presentation of the model would benefit from further development and clarifications to ensure that a wider audience can easily understand it. This applies to the whole modelling section but especially to the section about bias-variance analysis.

Thanks for this suggestion. We have added a paragraph to the section of the methods in which we describe the bias-variance analysis (p. 14):

“Intuitively, bias measures how much a participant’s actions deviate from the optimal policy. A bias of 0 indicates that the participant always follows the optimal policy. Positive values indicate that the participant is responding Go more frequently than optimal. The variance measures how much a participant’s actions deviate from her average policy. A variance of 0 indicates that the participant always gives the same response.”

We have also added two paragraphs to the Results section providing a more intuitive explanation of the modeling (p. 4-5):

“We developed two models of behavior on this task (see Methods for details). Both models consist of two sub-components, a Pavlovian learning system and an instrumental learning system (Fig. 3). The Pavlovian system acquires reward expectations for each stimulus, and converts these expectations into action values by promoting Go responses to cues in proportion to their expected reward. The instrumental system acquires reward expectations for each stimulus-action combination, and converts these into action values by promoting Go responses to stimuli in proportion to their expected reward for Go relative to No-Go. The learning rules for both systems are the same.

The Pavlovian and instrumental values are integrated linearly according to a weighting parameter. The two models differ in terms of how the weighting parameter changes as a function of experience. In the fixed model, the weighting parameter is held constant, treated as a free parameter that we fit to behavior. In the adaptive model, the weighting parameter is updated after each trial based on the relative predictive ability of each system. Thus, the weight is not a free parameter in the adaptive model, but is instead determined endogenously by each participant’s experience in the task.”

4) The authors assume that the instrumental learning controller is unbiased. Recent computational and experimental work suggest that biased instrumental learning mechanisms that reflects dopamine dependent learning mechanisms in the basal ganglia (Swart et al 2017, eLife). The authors should consider this work and its implications for their model in the discussion.

Thanks for raising this issue. We now include a new section on instrumental learning biases in the results (p. 9):

“Recent work by Swart and colleagues has shown that the Go bias is at least partly accounted for by instrumental learning biases. In particular, subjects in their study tended to learn more quickly from rewarded trials compared to non-rewarded trials, whereas they learned more slowly from punished trials compared to non-punished trials. This instrumental learning bias causes Go responses to appear more attractive when correct actions yield reward, and less attractive when correct actions yield avoidance of punishment.

Both the adaptive and fixed models will produce instrumental learning biases whenever the initial expected reward estimate θ_0 for the instrumental system deviates from 0.5. When $\theta_0 < 0.5$, the models will learn more from reward than from non-reward because the prediction error δ will be larger in the former case. Consistent with the findings of Swart and colleagues,

we found that the median parameter estimate for θ_0 was 0.35 in Experiment 1 and 0.30 in Experiment 2, both significantly below 0.5 ($p < 0.001$, t-test). Importantly, the parameter estimate in Experiment 1 did not differ significantly for subjects in the Low and High control conditions ($p = 0.49$, two-sample t-test), and thus cannot explain our key experimental finding that the Go bias differs as a function of controllability.”

5) reference 28 appears to be incomplete

Fixed.

Reviewer #2

1) If what is determining the balance is controllability, then we expect to see the same effects when in HC condition the probabilities are 20(GO)-80(No-GO)—the controllability in this condition is the same as the condition reported in the paper and therefore they should provide same Go Bias. I find this important to test this condition in order to confirm that the effects are purely due to controllability and not due to the imbalance in the dataset in HC trials.

We now show the full data across conditions and stimuli in Fig. S1. As shown in that figure, the go responding to the high control decoy is similar to responding to the Go-to-Win stimulus. Note that the reason we compare the Go and NoGo stimuli across conditions is because they are equated and thus this constitutes a controlled comparison.

2) I found it kind of odd that the weight parameter is determined based on which model provides the best fit to data. The adaptive model basically gives more weight to the system which can predict the data better. I think it is always the case that when we give more weight to the system which explained the data better (in a hybrid structure), the overall fit will be better than a hybrid structure which gives a fixed weight to each system. As such, the better fit of the adaptive model (compared to the fixed model) does not provide new information. In order to avoid this circular argument, the weight parameter I think should be determined by another measure, e.g., directly through controllability, instead of indirectly through the model fits.

It is not the case that the adaptive model will always explain the data better. The adaptive model does not introduce additional statistical degrees of freedom; on the contrary, it actually has fewer parameters than the fixed model. In other words, the adaptive model is less, not more, complex than the fixed model, in the sense of modeling flexibility. The critical distinction here is between modeling flexibility (roughly, the number of free parameters that we, the experimenters, have available to adjust the fit of the model to the data) and the flexibility that is endogenous to the model (the ability of the model to adapt itself, which is independent of the degrees of freedom available to the experimenters).

3) From Figure 4 left panel, it seems that the differences between model fits is driven by the learning process of the subjects, and it is not purely reflecting the controllability of the environment. That is, in both conditions the instrumental model is able to provide the better fit to data but it needs more learning. It then bears the question of whether the

transient effects that differentiate controllability (early trials in Figure 4 left) are rooted in the Pavlovian and instrumental nature of the models (stimulus vs stimulus-action values tracking) or the transient effects are purely due to the specific model parameterisation used here. For example, would we see a similar pattern if a fixed learning rate is used instead of the one in Line 188? What if the instrumental system is parametrized to track the difference between values? Please clarify the generality of the results.

These are interesting suggestions, but we see a few issues with implementing them. One can't specify a fixed learning rate for the adaptive Bayesian model; it would no longer correspond to coherent probabilistic inference, and thus it would not be clear how to interpret the posterior over controllability. A similar issue would obtain if we parameterized the instrumental system to track the value difference.

It's worth noting that with a fixed learning rate and a fixed Pavlovian bias, we would not expect to see our empirical observation that the go bias disappears for later trials. It's also worth noting that the learning rate for the instrumental and Pavlovian controllers is different from the rate of learning for w . In fact, the posterior over w will continue to update even after the controller values have stopped updating.

4) Bias-variance terms in line 248 – The values of these two quantities shown in Fig 7 are very small. I suspect a division by the number of trials is required in these equations. The relationship between bias and variance here and the trade-off between the bias and variance of an estimator is not very clear. With two actions if the subject is biased towards one of them then naturally the variance will be lower. It would be useful if the authors expand what exactly is going to be estimated here (the optimal action?) and whether the variance and bias generated by the two models are inline the assumptions.

Thanks for pointing this out; we had omitted the division by number of trials.

We have expanded the explanation of the bias-variance analysis in the methods, which we hope addresses any ambiguities (p. 14):

“Intuitively, bias measures how much a participant’s actions deviate from the optimal policy. A bias of 0 indicates that the participant always follows the optimal policy. Positive values indicate that the participant is responding Go more frequently than optimal. The variance measures how much a participant’s actions deviate from her average policy. A variance of 0 indicates that the participant always gives the same response.”

The reviewer is correct that if a subject is biased towards one action then the variance will be lower. However, bias here refers specifically to the difference between the chosen and optimal actions.

Minor comments:

5) Please add the derivations of the equations, e.g., the equation below line 200 to supplementary materials (and please number the equations).

We now included derivations in the supplement and number all equations in the main text.

6) How the optimal action (a^*_n) was chosen in the 50-50 conditions (line 248)?

There is no optimal action for this condition, as we now clarify in the text.

7) Please show the performance of the subjects.

Please see Fig. S1.

8) Equations below line 248, it would be better to replace the summation index with 't'.

Done.

9) It might not be accurate to talk about “actions values” within a Pavlovian system (line 49), as learning the value of actions (as opposed to learning the value of stimuli) is more related to the instrumental processes.

We agree that it is somewhat odd to talk about Pavlovian action values, but this is precisely what all the research on Pavlovian-instrumental actions implies: the Pavlovian system can impinge on action selection, and hence it can be described as de facto affecting action values.

10) Please show Figure 4 (left) for each trial type.

The controllability (w) is a global parameter, not specific to a trial type. We are primarily interested in how this parameter changes as a function of the experimental condition (high vs. low control).

Reviewer #3

1) It is stated that most previous work assumes the Pavlovian bias to be a fixed trait. This is not the case. Consider the extensive attempts to assess the effects of psychoactive drugs, but also other states such as stress, threat of shock and ‘cognitive control’ on precisely this bias.

Whether or not Pavlovian bias is a fixed trait or dynamic process has not yet been determined, with even recent work presenting these two possibilities (for example, see: Moutoussis et al., 2018). As such, we prefer to leave the wording as-is, since the goal of this study is to put forth one way in which Pavlovian biases might vary within individuals.

2) The number of excluded subjects (e.g. 91 out of 189 in Exp1) is enormous, even for an MTurk study. What is going on? Can their behavior be characterized (e.g. in terms of its consistency with an exclusive Pavlovian bias strategy, or a simple winstay/loseshift strategy) and reported as such?

It turned out that in fact this suspicious number reflected the fact that there was a bug in our experiment code, necessitating that we rerun Experiment 1. The proportion of excluded subjects is still rather high (around 30%) but not completely implausible for an MTurk study with stringent performance criteria.

3) I was puzzled by the lack of a punishment manipulation for 2 reasons:

First, doesn't this imply that any of the effects might reflect modulation of a nonspecific Go bias rather than specifically a Pavlovian bias? And might this represent a spillover from a generally increased go bias for the 80% stimulus? Second, the discussion highlights implications of the results for depression and learned helplessness, which concerns the aversive rather than appetitive domain.

Thanks for this suggestion. We agree that a punishment condition would be helpful in establishing the generality of our conclusions. However, it is not crucial for addressing the points raised by the reviewer. We now address the possibility of a non-specific Go bias on p. 10:

“Our results cannot be explained by a non-specific Go bias, whereby Go responses are rewarded more in the High control condition, inducing an overall tendency to produce Go responses. This would in fact predict the opposite effect (stronger Go bias under high reward controllability), contrary to our experimental findings.”

With regard to implications for depression and learned helplessness, we now including additional discussion on p. 10:

“Although the learned helplessness literature has focused on uncontrollable punishments, there is also evidence that individuals with depression are less likely to experience illusions of control with rewards (Alloy & Abramson, 1982).”

4) Am I completely confused or is it the adaptive and not the fixed model that has 6 parameters, including the weighting parameter? (page 7)

It is the fixed model that has the extra (6th) parameter. This is because only in the fixed model is the weighting term fit as a free parameter. In the adaptive model, the weighting term is not fit to the data, but is instead derived using the Bayesian model averaging method described on pages 6-7.

5) Can the proportion of Go responses be plotted separately for the 75% go and the 25% nogo stimuli? This would enable the reader to assess whether the 80% stimulus reduced Go responding for the 25% nogo stimulus or increased it for the 75% go stimulus. Might the uncontrollable context elicit a less specific 'inertia' rather than a 'Pavlovian go bias'?

We now show this plot in Fig S1.

6) Can the estimated Pavlovian weight parameter be plotted across trials (as in the simulation Fig 4)? In fact, does it make sense to report all (predicted and estimated) parameter distributions?

We now show the weight dynamics for a representative subject in Fig 4 (the weight dynamics are too variable to give meaningful results when averaged across subjects).

We are not entirely sure what is meant here by “predicted and estimated” parameter distributions. All we have access to are the parameter estimates. But critically, the Pavlovian weight is not an estimated parameter --- we do not fit it directly to behavior. It is endogenized by the model, and hence is a function of the other parameters in the model.

Perhaps some bits of the paper are too concise.

(i) The reader would benefit from a clearer description of Pavlovian vs instrumental control, with reference to classical and operant conditioning.

We now elaborate on this point more in the beginning of the Introduction (p. 1):

“A longstanding distinction holds that a Pavlovian learning system controls behavioral responses based on stimulus-outcome relationships (independently of actions), whereas a separate instrumental learning system controls responses based on stimulus-action-outcome relationships. In violation of this strict dichotomy, Pavlovian processes are known to promote approach towards reward-predictive stimuli and avoidance of punishment-predictive stimuli, even when they produce maladaptive behavior.”

(ii) Moreover, the authors could help the non-Bayesian-expert reader (like me) by making more intuitive earlier in the methods how to think about the mechanism by which environmental (un)controllability is inferred and how the fixed model differs conceptually from the adaptive model.

We have added two paragraphs to the Results section providing a more intuitive explanation of the modeling (p. 4):

“We developed two models of behavior on this task (see Methods for details). Both models consist of two sub-components, a Pavlovian learning system and an instrumental learning system (Fig. 3). The Pavlovian system acquires reward expectations for each stimulus, and converts these expectations into action values by promoting Go responses to cues in proportion to their expected reward. The instrumental system acquires reward expectations for each stimulus-action combination, and converts these into action values by promoting Go responses to stimuli in proportion to their expected reward for Go relative to No-Go. The learning rules for both systems are the same.

The Pavlovian and instrumental values are integrated linearly according to a weighting parameter. The two models differ in terms of how the weighting parameter changes as a function of experience. In the fixed model, the weighting parameter is held constant, treated as a free parameter that we fit to behavior. In the adaptive model, the weighting parameter is updated after each trial based on the relative predictive ability of each system. Thus, the weight is not a free parameter in the adaptive model, but is instead determined endogenously by each participant’s experience in the task.”

(iii) I am missing details of the simulation procedure. Is it the adaptive model that is simulated?

Thank you for pointing this out. We have revised the Figure 4 caption to clarify that it is the adaptive model that is being simulated.

(iv) Figure 6 plots the Go bias as a function of time. Is this from the low control condition? What about the high control condition? Why not present data from the same analysis also for the within-subjects and note the order issue. Here wouldn’t a plot of the evolution of the weight parameter make sense?

Thanks for this suggestion. We have now updated Figure 6 and the text to show the results for both experiments, which come to the same conclusion.

(v) It is unclear what is the added explanatory value is of the bias-variance trade-off analysis. In fact, this compounds another question I had with regard to the statement that the current data provide evidence for a Bayesian model averaging model. What is the explanatory value of this framework if the Bayesian model averaging approach in the absence of evidence that it does a better job than alternative non-Bayesian approaches?

We of course cannot rule out all possible alternative non-Bayesian models. Our goal was more circumscribed: we wanted to rule out the most well-known model in which Pavlovian bias is treated as a fixed trait. In fact, although the original Guitart-Masip model was non-Bayesian, we developed a Bayesian version of his model in order to better equate the adaptive and fixed versions of the model (so that the only difference was whether the model infers which controller to trust).

If we've understood the reviewer correctly, the comment is pointing out that the bias-variance analysis may be consistent with non-Bayesian models. We agree. Our goal was not to rule out non-Bayesian models, but rather propose a specific Bayesian model that can capture the effect of controllability on Pavlovian bias. The bias-variance analysis provides a way of supporting our argument that the locus of this effect lies in a form of model comparison. Again, this could be implemented in a non-Bayesian way, and we don't rule that out.

Reviewers' Comments:

Reviewer #1:

Remarks to the Author:

The authors have done a good work addressing the comments raised by me and the other reviewers. However, I think that there are 3 remaining points that still need to be properly addressed:

- In relation to my first comment, the authors state in their response that the operational definition of the go bias is a difference in performance between go-to-win and no-go-to-win. This is very clear and answers my concern. However, in the manuscript, the definition reads: "Go bias, defined as the accuracy difference between Go and No-Go trials". This is less clear and may still confuse a potential reader, as it confused me in the previous round. I would suggest that the definition of Go bias is clarified in the main text.

- Regarding the second point I rose on the previous round, why don't the authors do their predictive check on the learning curves that they provide on figure 6? It seems to me that this would provide a simple and intuitive visual confirmation that the model is capturing the data without recurring to the model predictions to bin the data.

- I do not think that the discussion of the Swart et al paper added to the manuscript discuss the main finding of that paper. The authors refer to the Swart et al paper like this: "In particular, subjects in their study tended to learn more quickly from rewarded trials compared to non-rewarded trials, whereas they learned more slowly from punished trials compared to non-punished trials." I think that this is not right, as the key aspect of the instrumental bias in Swart et al is not a difference in learning between rewarded and non-rewarded trials but on the interaction between reward and action. Instead, it should read like this: "In particular, subjects in their study tended to learn more quickly from rewarded go trials compared to rewarded no-go trials, whereas they learned more slowly from punished no-go trials compared to punished go trials." This affects the subsequent line of argumentation added to the discussion in relation to this previous article.

Reviewer #2:

Remarks to the Author:

In this revision, the authors have made changes which have improved the manuscript, in particular, Figure S1. I went over the paper again, but I still have some of the concerns that I raised earlier:

1) In the decoy trials the authors use 80(GO)/20(NO GO) in HC condition. Alternatively, they could have used 20(GO)/80(NO GO) in HC condition, which has the same controllability properties of 80(GO)/20(NO GO), and therefore it should affect GO bias in the same way. Is there any reason that the authors chose to use 80(GO)/20(NO GO) condition only and did not test 20(GO)/80(NO GO) condition? If so, it would be good to add that to the paper. Otherwise, I find it important to test 20(GO)/80(NO GO) to make it solid that the current GO bias seen in the results is not because of the imbalance between GO and NO GO reward probabilities in HC decoy condition and it is purely due to controllability. For example, it could be the case that since subjects took more GO responses in the decoy trials of HC condition, they made less GO responses in GO-To-Win trials, to somehow balance the exploration between Go and NO GO actions over all the trial types. In any case, please comment on this design choice in the paper and the alternative explanations.

2) I am convinced by the response.

3) I am convinced that implementing the fixed learning rate is not straight forward in the current Bayesian formulation. But I was not able to follow why parameterising the instrumental system with a single action value is not possible. Please discuss this.

4) The new paragraph reads well, but as mentioned in my previous comment it is still unclear whether the two proposed models for instrumental and Pavlovian systems exhibit bias/variance properties that are shown here, i.e., Pavlovian model shows higher bias and lower variance and instrumental shows higher variance and lower bias. If the claim is that these two computational models have these properties -- which is the case according to the introduction -- then it is necessary to should it here.

Reviewer #3:

Remarks to the Author:

see attached

The rebuttal is succinct and to the point, and at times overly minimalist. I am generally satisfied with the rebuttal, but for some minor residual points.

My responses to the authors' rebuttal of my previous comments are indented. And accompanied by a

#R3

1) It is stated that most previous work assumes the Pavlovian bias to be a fixed trait. This is not the case. Consider the extensive attempts to assess the effects of psychoactive drugs, but also other states such as stress, threat of shock and 'cognitive control' on precisely this bias.

AU: Whether or not Pavlovian bias is a fixed trait or dynamic process has not yet been determined, with even recent work presenting these two possibilities (for example, see: Moutoussis et al., 2018). As such, we prefer to leave the wording as-is, since the goal of this study is to put forth one way in which Pavlovian biases might vary within individuals.

➤ I can see that the goal of this study is to propose one way in which biases might vary within individuals. However, it is wrong to state that the bias is assumed so far to be a fixed trait (even though the few formal modeling attempts do so). Please update.

2) The number of excluded subjects (e.g. 91 out of 189 in Exp1) is enormous, even for an MTurk study. What is going on? Can their behavior be characterized (e.g. in terms of its consistency with an exclusive Pavlovian bias strategy, or a simple winstay/loshift strategy) and reported as such?

AU: It turned out that in fact this suspicious number reflected the fact that there was a bug in our experiment code, necessitating that we rerun Experiment 1. The proportion of excluded subjects is still rather high (around 30%) but not completely implausible for an MTurk study with stringent performance criteria.

➤ This is a little worrying. In which other ways did the bug in experimental code affect the results? Moreover, please address my request to characterize and report the behavior of the excluded subjects.

3) I was puzzled by the lack of a punishment manipulation for 2 reasons: First, doesn't this imply that any of the effects might reflect modulation of a nonspecific Go bias rather than specifically a Pavlovian bias? And might this represent a spillover from a generally increased go bias for the 80% stimulus? Second, the discussion highlights implications of the results for depression and learned helplessness, which concerns the aversive rather than appetitive domain.

AU: Thanks for this suggestion. We agree that a punishment condition would be helpful in establishing the generality of our conclusions. However, it is not crucial for addressing the points raised by the reviewer. We now address the possibility of a non-specific Go bias on p. 10: "Our results cannot be explained by a non-specific Go bias, whereby Go responses are rewarded more in the High control condition, inducing an overall tendency to produce Go responses. This would in fact predict the opposite effect (stronger Go bias under high reward controllability), contrary to our experimental findings."

- The nonspecific effect might pan out in the opposite direction, though, with people making unmodeled assumptions about a certain minimal number of required go-responses to be divided across all trial-types. The addition of a punishment condition would provide a more specific test, and it seems worth adding this point to the discussion.

5) Can the proportion of Go responses be plotted separately for the 75% go and the 25% nogo stimuli? This would enable the reader to assess whether the 80% stimulus reduced Go responding for the 25% nogo stimulus or increased it for the 75% go stimulus. Might the uncontrollable context elicit a less specific ‘inertia’ rather than a ‘Pavlovian go bias’?

AU: We now show this plot in Fig S1.

- Why not actually provide an answer in this rebuttal letter rather than simply referring to the figure? This figure demonstrates the subtlety of the effect. In fact, it is not observable in this visualization. What would be more informative is to plot the effect of control on $p(\text{Go})$ as a function of trial.

6) Can the estimated Pavlovian weight parameter be plotted across trials (as in the simulation Fig 4)? In fact, does it make sense to report all (predicted and estimated) parameter distributions?

AU: We now show the weight dynamics for a representative subject in Fig 4 (the weight dynamics are too variable to give meaningful results when averaged across subjects).

- But shouldn't there be systematic fluctuations in this weight parameter corresponding to the decoy reward contingency?

AU: We are not entirely sure what is meant here by “predicted and estimated” parameter distributions. All we have access to are the parameter estimates. But critically, the Pavlovian weight is not an estimated parameter --- we do not fit it directly to behavior. It is endogenized by the model, and hence is a function of the other parameters in the model.

- I was referring to the other parameters in the model

(ii) Moreover, the authors could help the non-Bayesian-expert reader (like me) by making more intuitive earlier in the methods how to think about the mechanism by which environmental (un)controllability is inferred and how the fixed model differs conceptually from the adaptive model.

Au: We have added two paragraphs to the Results section providing a more intuitive explanation of the modeling (p. 4): “We developed two models of behavior on this task (see Methods for details). Both models consist of two sub-components, a Pavlovian learning system and an instrumental learning system (Fig. 3). The Pavlovian system acquires reward expectations for each stimulus, and converts these expectations into action values by promoting Go responses to cues in proportion to their expected reward. The instrumental system acquires reward expectations for each stimulus-action combination, and converts these into action values by promoting Go responses to stimuli in proportion to their expected reward for Go relative to No-Go. The learning rules for both systems are the same.

The Pavlovian and instrumental values are integrated linearly according to a weighting parameter. The two models differ in terms of how the weighting parameter changes as a

function of experience. In the fixed model, the weighting parameter is held constant, treated as a free parameter that we fit to behavior. In the adaptive model, the weighting parameter is updated after each trial based on the relative predictive ability of each system. Thus, the weight is not a free parameter in the adaptive model, but is instead determined endogenously by each participant's experience in the task."

➤ Okay thanks

(iii) I am missing details of the simulation procedure. Is it the adaptive model that is simulated?

AU: Thank you for pointing this out. We have revised the Figure 4 caption to clarify that it is the adaptive model that is being simulated.

(iv) Figure 6 plots the Go bias as a function of time. Is this from the low control condition? What about the high control condition? Why not present data from the same analysis also for the within-subjs exp and note the order issue. Here wouldn't a plot of the evolution of the weight parameter make sense?

AU: Thanks for this suggestion. We have now updated Figure 6 and the text to show the results for both experiments, which come to the same conclusion.

➤ What about a plot of the evolution of the weight parameter across trials?

(v) It is unclear what is the added explanatory value of the bias-variance trade-off analysis. In fact, this compounds another question I had with regard to the statement that the current data provide evidence for a Bayesian model averaging model. What is the explanatory value of this framework if the Bayesian model averaging approach in the absence of evidence that it does a better job than alternative non-Bayesian approaches?

AU: We of course cannot rule out all possible alternative non-Bayesian models. Our goal was more circumscribed: we wanted to rule out the most well-known model in which Pavlovian bias is treated as a fixed trait. In fact, although the original Guitart-Masip model was non-Bayesian, we developed a Bayesian version of his model in order to better equate the adaptive and fixed versions of the model (so that the only difference was whether the model infers which controller to trust).

AU: If we've understood the reviewer correctly, the comment is pointing out that the bias-variance analysis may be consistent with non-Bayesian models. We agree. Our goal was not to rule out non-Bayesian models, but rather propose a specific Bayesian model that can capture the effect of controllability on Pavlovian bias. The bias-variance analysis provides a way of supporting our argument that the locus of this effect lies in a form of model comparison. Again, this could be implemented in a non-Bayesian way, and we don't rule that out.

➤ Okay but then it seems the conclusion that the current data provide evidence for a Bayesian account is not so well put

Computational Cognitive Neuroscience Lab
Harvard University
52 Oxford Street
Cambridge, MA 02138

Reviewer #1

The authors have done a good work addressing the comments raised by me and the other reviewers. However, I think that there are 3 remaining points that still need to be properly addressed:

- 1) In relation to my first comment, the authors state in their response that the operational definition of the go bias is a difference in performance between go-to-win and no-go-to-win. This is very clear and answers my concern. However, in the manuscript, the definition reads: "Go bias, defined as the accuracy difference between Go and No-Go trials". This is less clear and may still confuse a potential reader, as it confused me in the previous round. I would suggest that the definition of Go bias is clarified in the main text.

We have now amended this definition to the *"difference in accuracy on Go-to-Win and No-Go-to-Win trials"*.

- 2) Regarding the second point I rose on the previous round, why don't the authors do their predictive check on the learning curves that they provide on figure 6? It seems to me that this would provide a simple and intuitive visual confirmation that the model is capturing the data without recurring to the model predictions to bin the data.

Thanks for this suggestion, we've now updated Fig. 6 to also show the model predictions.

- 3) I do not think that the discussion of the Swart et al paper added to the manuscript discuss the main finding of that paper. The authors refer to the Swart et al paper like this: "In particular, subjects in their study tended to learn more quickly from rewarded trials compared to non-rewarded trials, whereas they learned more slowly from punished trials compared to non-punished trials." I think that this is not right, as the key aspect of the instrumental bias in Swart et al is not a difference in learning between rewarded and non-rewarded trials but on the interaction between reward and action. Instead, it should read like this: "In particular, subjects in their study tended to learn more quickly from rewarded go trials compared to rewarded no-go trials, whereas they learned more slowly from punished no-go trials compared to punished go trials." This affects the subsequent line of argumentation added to the discussion in relation to this previous article.

Thanks for pointing this out. We have changed the discussion of the Swart study as suggested. Most importantly, we think that this phenomenon is actually not captured by our model. We have therefore removed the data analysis and shifted this to the Discussion (p. 10):

"Recent work by Swart and colleagues complicates this picture by showing that the Go bias is at least partly accounted for by instrumental learning biases. In particular, subjects in their study tended to learn more quickly from rewarded Go trials compared to rewarded No-Go trials, whereas they learned more slowly from punished No-Go trials compared to punished Go trials."

This instrumental learning bias causes Go responses to appear more attractive when correct actions yield reward, and less attractive when correct actions yield avoidance of punishment. This phenomenon is not accounted for by our modeling framework."

Reviewer #2

In this revision, the authors have made changes which have improved the manuscript, in particular, Figure S1. I went over the paper again, but I still have some of the concerns that I raised earlier:

- 1) In the decoy trials the authors use 80(GO)/20(NO GO) in HC condition. Alternatively, they could have used 20(GO)/80(NO GO) in HC condition, which has the same controllability properties of 80(GO)/20(NO GO), and therefore it should affect GO bias in the same way. Is there any reason that the authors chose to use 80(GO)/20(NO GO) condition only and did not test 20(GO)/80(NO GO) condition? If so, it would be good to add that to the paper. Otherwise, I find it important to test 20(GO)/80(NO GO) to make it solid that the current GO bias seen in the results is not because of the imbalance between GO and NO GO reward probabilities in HC decoy condition and it is purely due to controllability. For example, it could be the case that since subjects took more GO responses in the decoy trials of HC condition, they made less GO responses in GO-To-Win trials, to somehow balance the exploration between Go and NO GO actions over all the trial types. In any case, please comment on this design choice in the paper and the alternative explanations.

We made this choice deliberately, because it allowed us to rule out an alternative hypothesis, which we explain on p. 9: *"Our results cannot be explained by a non-specific Go bias, whereby Go responses are rewarded more in the High control condition, inducing an overall tendency to produce Go responses. This would in fact predict the opposite effect (stronger Go bias under high reward controllability), contrary to our experimental findings."*

The idea that subjects are somehow trying to balance their exploration across stimuli is intriguing, but this strikes us as a pretty ad hoc hypothesis, not motivated by any theories that we are aware of. In any case, we don't believe it can fit our data, since it would predict that the probability of producing a Go response on Go-to-Win trials should be inversely correlated with the probability of producing a Go response on decoy trials, but this correlation is not significant.

- 2) I am convinced that implementing the fixed learning rate is not straight forward in the current Bayesian formulation. But I was not able to follow why parameterising the instrumental system with a single action value is not possible. Please discuss this.

Sorry, the reviewer is correct that such a model is not necessarily incompatible with a Bayesian formulation. We are, however, struggling to understand why this would be an interesting alternative model to consider. Our choice of models was strongly constrained by the prior modeling tradition that started with Guitart-Masip et al. (2012), and which has proven to be very successful across numerous papers using this task. So we felt it would be appropriate to keep as close as possible to this original modeling framework.

- 3) The new paragraph reads well, but as mentioned in my previous comment it is still unclear whether the two proposed models for instrumental and Pavlovian systems exhibit bias/variance properties that are shown here, i.e., Pavlovian model shows higher bias and lower variance and instrumental shows higher variance and lower bias. If the claim is that these two computational models have these properties -- which is the case according to the introduction -- then it is necessary to should it here.

This has to be true mathematically (it's not an empirical claim), because the Pavlovian model has a strictly stronger inductive bias than the instrumental model. One problem with presenting the bias/variance analysis for the models side-by-side with the analysis for the data is that the variance is scaled differently for the models: they give a probabilistic output, and therefore will generally have lower variance than the data (which are binary) even though they show the same relative pattern when comparing controllability conditions.

Reviewer #3

The rebuttal is succinct and to the point, and at times overly minimalist. I am generally satisfied with the rebuttal, but for some minor residual points.

My responses to the authors' rebuttal of my previous comments are indented (NOTE: the authors have also underlined these new reviewer responses for ease of reading).

- 1) It is stated that most previous work assumes the Pavlovian bias to be a fixed trait. This is not the case. Consider the extensive attempts to assess the effects of psychoactive drugs, but also other states such as stress, threat of shock and 'cognitive control' on precisely this bias.

AU: Whether or not Pavlovian bias is a fixed trait or dynamic process has not yet been determined, with even recent work presenting these two possibilities (for example, see: Moutoussis et al., 2018). As such, we prefer to leave the wording as-is, since the goal of this study is to put forth one way in which Pavlovian biases might vary within individuals.

I can see that the goal of this study is to propose one way in which biases might vary within individuals. However, it is wrong to state that the bias is assumed so far to be a fixed trait (even though the few formal modeling attempts do so). Please update.

We have now removed all references to the "fixed trait" point.

- 2) The number of excluded subjects (e.g. 91 out of 189 in Exp1) is enormous, even for an MTurk study. What is going on? Can their behavior be characterized (e.g. in terms of its consistency with an exclusive Pavlovian bias strategy, or a simple winstay/loseshift strategy) and reported as such?

AU: It turned out that in fact this suspicious number reflected the fact that there was a bug in our experiment code, necessitating that we rerun Experiment 1. The proportion of excluded subjects is still rather high (around 30%) but not completely implausible for an MTurk study with stringent performance criteria.

This is a little worrying. In which other ways did the bug in experimental code affect the results? Moreover, please address my request to characterize and report the behavior of the excluded subjects.

The bug invalidated the experimental results because the experimental manipulation was not implemented properly. We fixed this error in the code and collected new data.

In the supplement (Fig S2), we now show the Go bias for both experiments without subject exclusions. In brief, there is a Go bias for both experiments, but no effect of controllability when we include all collected subject data. However, note that we chose these selection criteria prior to analyzing the Go bias. Many studies of the Mechanical Turk subject population have emphasized the need to exclude subjects (typically more than what is standard in a laboratory task) in order to have interpretable data.

3) I was puzzled by the lack of a punishment manipulation for 2 reasons: First, doesn't this imply that any of the effects might reflect modulation of a nonspecific Go bias rather than specifically a Pavlovian bias? And might this represent a spillover from a generally increased go bias for the 80% stimulus? Second, the discussion highlights implications of the results for depression and learned helplessness, which concerns the aversive rather than appetitive domain.

AU: Thanks for this suggestion. We agree that a punishment condition would be helpful in establishing the generality of our conclusions. However, it is not crucial for addressing the points raised by the reviewer. We now address the possibility of a non-specific Go bias on p. 10: "Our results cannot be explained by a non-specific Go bias, whereby Go responses are rewarded more in the High control condition, inducing an overall tendency to produce Go responses. This would in fact predict the opposite effect (stronger Go bias under high reward controllability), contrary to our experimental findings."

The nonspecific effect might pan out in the opposite direction, though, with people making unmodeled assumptions about a certain minimal number of required go-responses to be divided across all trial-types. The addition of a punishment condition would provide a more specific test, and it seems worth adding this point to the discussion.

Thanks for the additional information. We have added this point to the Discussion (p. 9): "*Even stronger evidence against a non-specific Go bias would be provided by a version of the experiment in which participants must make Go/No-Go responses to avoid punishment.*"

4) Can the proportion of Go responses be plotted separately for the 75% go and the 25% nogo stimuli? This would enable the reader to assess whether the 80% stimulus reduced Go responding for the 25% nogo stimulus or increased it for the 75% go stimulus. Might the uncontrollable context elicit a less specific 'inertia' rather than a 'Pavlovian go bias'?

AU: We now show this plot in Fig S1.

Why not actually provide an answer in this rebuttal letter rather than simply referring to the figure? This figure demonstrates the subtlety of the effect. In fact, it is not observable in this visualization. What would be more informative is to plot the effect of control on p(Go) as a function of trial.

Thanks for the clarification. We have now replaced Fig S1 with a reorganized plot as suggested. This shows that the effect seems to be slightly stronger on the Go-to-Win cue.

5) Can the estimated Pavlovian weight parameter be plotted across trials (as in the simulation Fig 4)? In fact, does it make sense to report all (predicted and estimated) parameter distributions?

AU: We now show the weight dynamics for a representative subject in Fig 4 (the weight dynamics are too variable to give meaningful results when averaged across subjects).

But shouldn't there be systematic fluctuations in this weight parameter corresponding to the decoy reward contingency?

Yes, but the time series representation is too noisy to discern this.

AU: We are not entirely sure what is meant here by "predicted and estimated" parameter distributions. All we have access to are the parameter estimates. But critically, the Pavlovian weight is not an estimated parameter --- we do not fit it directly to behavior. It is endogenized by the model, and hence is a function of the other parameters in the model.

I was referring to the other parameters in the model

Thanks for clarifying. We now report the average parameter estimates in Table S1.

6) Figure 6 plots the Go bias as a function of time. Is this from the low control condition? What about the high control condition? Why not present data from the same analysis also for the within-subj exp and note the order issue. Here wouldn't a plot of the evolution of the weight parameter make sense?

AU: Thanks for this suggestion. We have now updated Figure 6 and the text to show the results for both experiments, which come to the same conclusion.

What about a plot of the evolution of the weight parameter across trials?

We did explore plotting this originally, but the problem is that there is a huge amount of between-subject variability in the weight trajectories, and this obscured any patterns. This is why we opted to plot a representative subject in Fig 4.

7) It is unclear what is the added explanatory value is of the bias-variance trade-off analysis. In fact, this compounds another question I had with regard to the statement that the current data provide evidence for a Bayesian model averaging model. What is the explanatory value of this framework if the Bayesian model averaging approach in the absence of evidence that it does a better job than alternative non-Bayesian approaches?

AU: We of course cannot rule out all possible alternative non-Bayesian models. Our goal was more circumscribed: we wanted to rule out the most well-known model in which Pavlovian bias is treated as a fixed trait. In fact, although the original Guitart-Masip model was non-Bayesian,

we developed a Bayesian version of his model in order to better equate the adaptive and fixed versions of the model (so that the only difference was whether the model infers which controller to trust).

AU: If we've understood the reviewer correctly, the comment is pointing out that the bias-variance analysis may be consistent with non-Bayesian models. We agree. Our goal was not to rule out non-Bayesian models, but rather propose a specific Bayesian model that can capture the effect of controllability on Pavlovian bias. The bias-variance analysis provides a way of supporting our argument that the locus of this effect lies in a form of model comparison. Again, this could be implemented in a non-Bayesian way, and we don't rule that out.

Okay but then it seems the conclusion that the current data provide evidence for a Bayesian account is not so well put

To be more conservative, we have changed "provide support for" to "provide evidence consistent with" a Bayesian model averaging theory.

Reviewers' Comments:

Reviewer #1:

Remarks to the Author:

The authors have successfully addressed my remaining concerns.

Reviewer #2:

Remarks to the Author:

Thank you for clarifying the first point -- I am convinced that the choice of 80/20 is appropriate here and a nice design choice.

About points 2 and 3, I still have the same concerns that I raised in the last two revisions -- it is unclear whether the points mentioned in the introduction about the complexity/bias-variance (Figure 1) hold for the computational models suggested here. It then questions whether the patterns of results (say Fig. 4) are rooted in the complexities of the models or they are just a consequence of the specific parameterizations used here -- as we do not know for the models used here which one is more complex or has a higher bias/variance. I suggested using an alternative parametrisation and showing that the results still hold, or alternatively showing that these properties of the systems are correct empirically (I am not sure why it is not possible to draw binary samples from the models to compare with the data), or the authors might have something else in mind to try. I might have misunderstood the logic of the paper, but as it stands I struggle to follow the links between the arguments in the introduction (Figure 1) and the actual computational models used here for each system.

Reviewer #3:

Remarks to the Author:

I have no further comments

Computational Cognitive Neuroscience Lab
Harvard University
52 Oxford Street
Cambridge, MA 02138

Reviewer #1

Thank you for clarifying the first point -- I am convinced that the choice of 80/20 is appropriate here and a nice design choice.

About points 2 and 3, I still have the same concerns that I raised in the last two revisions -- it is unclear whether the points mentioned in the introduction about the complexity/bias-variance (Figure 1) hold for the computational models suggested here. It then questions whether the patterns of results (say Fig. 4) are rooted in the complexities of the models or they are just a consequence of the specific parameterizations used here -- as we do not know for the models used here which one is more complex or has a higher bias/variance. I suggested using an alternative parametrisation and showing that the results still hold, or alternatively showing that these properties of the systems are correct empirically (I am not sure why it is not possible to draw binary samples from the models to compare with the data), or the authors might have something else in mind to try. I might have misunderstood the logic of the paper, but as it stands I struggle to follow the links between the arguments in the introduction (Figure 1) and the actual computational models used here for each system.

Thank you for clarifying this concern. To better demonstrate the connection between the bias-variance framework and the proposed models, we have added Figure 3 to the Supplement (and shown below). Figure 3 is a simulation illustrating that a purely Pavlovian agent has higher bias and lower variance compared to a purely instrumental agent.

Supplementary Figure 3. Bias-variance analysis for purely Pavlovian ($w = 1$) and purely instrumental ($w = 0$) agents.